# L-Arginine Modulates Neonatal Leukocyte Recruitment in a Gestational Age-Dependent Manner

**DOI:** 10.3390/jcm9092772

**Published:** 2020-08-27

**Authors:** Raphaela Fitterer, Trim Lajqi, Simon Alexander Kranig, Maylis Braun, Nicole Theissig, Navina Kuss, Johannes Pöschl, David Frommhold, Hannes Hudalla

**Affiliations:** 1Department of Neonatology, Heidelberg University Children’s Hospital, 69120 Heidelberg, Germany; raphaela.fitterer@med.uni-heidelberg.de (R.F.); trim.lajqi@med.uni-heidelberg.de (T.L.); simon.kranig@med.uni-heidelberg.de (S.A.K.); maylis.braun@med.uni-heidelberg.de (M.B.); nicole_theissig@yahoo.de (N.T.); navina.kuss@med.uni-heidelberg.de (N.K.); johannes.poeschl@med.uni-heidelberg.de (J.P.); 2Klinik für Kinderheilkunde und Jugendmedizin, 87700 Memmingen, Germany; David.Frommhold@klinikum-memmingen.de

**Keywords:** neutrophils, leukocyte recruitment, L-arginine, newborn, innate immunity

## Abstract

(1) Background: L-arginine is a complex modulator of immune functions, and its levels are known to decrease under septic conditions. L-arginine may suppress leukocyte recruitment in vivo; however, little is known about the gestational age-specific effects of L-arginine on leukocyte recruitment in preterm infants. We now asked whether L-arginine alters leukocyte recruitment in preterm and term neonates. (2) Methods: Leukocytes were isolated from preterm (28 + 0 to 32 + 6 weeks of gestation) and term (>37 weeks of gestation) newborns as well as from healthy adults. After incubation with 10 µg/mL L-arginine, we assessed leukocyte rolling and adhesion in dynamic microflow chamber experiments and leukocyte transmigration in fluorescence assays. In addition, we measured the expression of inducible nitric oxide synthase (iNOS) and Arginase 1 (Arg-1) in neutrophils by flow cytometry. (3) Results: Leukocyte rolling, adhesion, and transmigration increased with gestational age. Leukocyte rolling, adhesion, and transmigration were decreased by L-arginine in term-born infants and adults. Preterm leukocytes showed no change in recruitment upon L-arginine exposure. Leukocyte adhesion after L-arginine exposure reached similar levels among all groups. In line, the expression of iNOS and Arg-1 was similar in all three age groups. (4) Conclusion: L-arginine dampens the ex vivo recruitment capacity of leukocytes from term-born infants, whereas no effect was seen in premature infants. As levels of iNOS and Arg-1 in neutrophils remain ontogenetically unchanged, the anti-inflammatory effect of L-arginine on the leukocyte recruitment cascade needs further investigation. These results add to the controversial debate of L-arginine supplementation in premature infants in sepsis.

## 1. Introduction

The high susceptibility to infections and sepsis in preterm infants remains a major problem in neonatology, leading to high morbidity and mortality [1]. Especially among very premature infants, the immaturity of the immune system contributes greatly to adverse outcome [2,3]. Neutrophils are the first responders in bacterial sepsis, and the leukocyte recruitment cascade initiates the response of the innate immune system [4,5]. Circulating leukocytes are first captured and start rolling along the endothelial layer [6]. The activation of β2 integrins such as LFA-1 (CD11a/CD18) and Mac-1 (CD11b/CD18) leads to binding with different endothelial ligands such as intercellular adhesion molecule 1 (ICAM-1), resulting in firm adhesion to the inflamed endothelium, which is followed by transmigration along a chemokine gradient [7]. Nussbaum et al. found that the fetal leukocyte recruitment in preterm infants is heavily impaired and undergoes a maturation process during fetal development [8]. In line, the expression of adhesion molecules such as P-selectin or ICAM-1 and the chemokine IL-8 increase with gestational age [9,10]. This ontogenetic maturation may partially account for the high susceptibility to sepsis in preterm infants [11].

L-arginine, a conditionally essential amino acid for the fetus and neonate, is known to be involved in the regulation of various immune functions. For example, L-arginine limitation dampens Toll-like receptor 4 (TLR-4) signaling in macrophages, and the supplementation of L-arginine can restore TNF-α production in response to lipopolysaccharide (LPS) [12]. Polymorphonuclear leukocytes (PMNs) express Arginase-1 (Arg-1) and inducible nitric oxide synthase (iNOS), which both catabolize L-arginine. Interestingly, it was shown that L-arginine depletion through neutrophilic arginase secretion may locally suppress T-cell functions, and restoring L-arginine levels had a pro-inflammatory effect [13]. By a similar mechanism, human natural-killer (NK) cells are suppressed by PMNs through L-arginine depletion [14]. On the other hand, L-arginine exerts anti-inflammatory effects on PMNs by reducing leukocyte activation and recruitment [15], partially through the downregulation of ICAM-1 on endothelial cells [16]. As L-arginine levels are lower in preterm infants [17,18], supplementation is common practice on neonatal intensive care units. Especially during sepsis, L-arginine levels rapidly drop, defining sepsis as an “arginine-deprived state” [19]. Immune modulation during sepsis needs to be stage-dependent, as early inflammation in newborns requires immune stimulation, whilst over-activation may induce a systemic inflammatory response syndrome (SIRS) with often fatal consequences. An example for successful immune modulation with L-arginine is the supplementation of low plasma levels during necrotizing enterocolitis (NEC) [20], which has been shown to reduce the incidence and morbidity in clinical trials and animal studies [21,22,23,24]. Our study asked whether gestational age alters the effect of L-arginine supplementation on the leukocyte recruitment cascade. In order to eliminate systemic pro- or anti-inflammatory effects of L-arginine, we used ex vivo flow chamber assays to assess peripheral PMNs from premature infants, term-born infants, and adults.

## 2. Materials and Methods

### 2.1. Study Population and Sample Collection

Infants of the “Priming Immunity at the Beginning of Life” (PRIMAL) study cohort [25] born between 28 + 0 and 32 + 6 weeks of gestation were recruited for the group of preterm infants. Term-born infants (≥37 + 0 weeks of gestation) were enrolled after delivery. Children showing fetal malformations were excluded from the study. First, 1 mL of blood was collected within the first day of life. The preterm study cohort was followed up with a second blood draw at 28 days after enrollment, as per the PRIMAL study protocol [25]. For adult controls, 5 mL of peripheral venous blood was obtained from healthy volunteers. For anticoagulation, standard blood collection tubes (Sarstedt Coagulation9NC) with trisodiumcitrate were used. The following clinical data were collected from electronic records: gestational age (GA) in weeks, 5 and 10 min APGAR (appearance, pulse, grimace, activity und respiration) score, gender, head circumference in cm, weight in grams, length in cm, breastfeeds (full or partial), rupture of membranes at birth, and premature rupture of membranes (PROM, >18 h before birth), treatment with antibiotics of the newborn, and birth mode (vaginal or C-section). Furthermore, the following laboratory parameters were collected: arterial cord pH, C-reactive protein (CRP) in mg/L, total white blood cell count (WBC) per nL, and the quotient of immature to total neutrophils (IT, considered elevated above 0.25).

### 2.2. Compliance with Ethical Standards

Informed, written consent for enrollment in the study and the publication of findings was obtained from all guardians or adult volunteers, and the study was approved by the local Medical Ethical Committee of Heidelberg University (name of ethics committee: Ethikkommission Medizinische Fakultät Heidelberg: project identification code: S-603/2013, date of approval: 04/04/2014; and: project identification code: S-168/2018, date of approval: 03/22/2018). All procedures performed were in accordance with the ethical standards and data safety of the institutional and national research committee and with the 1964 Helsinki declaration and its later amendments.

### 2.3. Isolation of Polymorphonuclear Leukocytes (PMNs)

PMNs were isolated as previously described [11]. In brief, whole blood was separated by density gradient (LSM 1077; PAA Laboratories GmbH, Coelbe, Germany) centrifugation (1200× *g*, 20 min, 4 °C). The resulting erythrocyte–granulocyte pellet was washed twice in Dulbecco’s PBS (1×, without Ca++ and Mg ++; Invitrogen GmbH, Darmstadt, Germany), followed by hypotonic erythrocyte lysis (0.15 M NH_4_Cl, 0.01 M NaHCO_3_, 0.001 M EDTA, in aqua ad injectabilia for 7 min in the dark at room temperature). PMNs were washed twice and counted in a Neubauer chamber (Bright-line R, Hausser Scientific Horsham, PA, USA). The purity of the isolated cell suspension was validated by cytomorphologic analysis via flow cytometry using standard leukocyte markers as previously described [11]. For experiments, PMNs were counted and diluted to reach a working suspension of 1 × 10^6^ cells/mL.

### 2.4. Flow Chamber Experiments

Leukocyte rolling and adhesion was determined by dynamic microflow chamber experiments in a novel setup due to limited blood volumes from preterm infants. Microglass capillaries (Ibidi µ-Slides VI 0.1; Sarstedt, Nümbrecht, Germany) were used to acquire high-resolution microscopy of cells under shear stress. To resemble standardized blood flow conditions, a high-precision perfusion pump (Harvard Instruments, March-Hugstetten, Germany) was used. As shown in the formula below, the flow rate in a rectangular channel is dependent on the viscosity and the shear stress of the perfused medium. In order to determine the flow rate to achieve a wall shear stress of 1 dyn/cm^2^, we first measured the dynamic viscosity of our cell suspension at room temperature using Rheodyn SSD as described previously [26]. With a viscosity of 0.01535 dyn·s/cm^2^, the flow rate was determined to be 8.87 µL /min in our setup.

Flow rate = Shear Stress/(Viscosity × 10.7)
$$\tau \left[\frac{dyn}{cm^{2}}\right]=\eta \left[\frac{dyn\cdot s}{cm^{2}}\right]\times 10.7\times \Phi [\frac{\mu L}{min}]$$

η = Viskosity; Φ = Flow rate, τ = Shear stress.

Then, capillaries were coated to imitate the endothelial layer. In previous experiments, coating with P-selectin 4 μg/mL (ADP3050, Bio-Techne GmbH, Wiesbaden, Germany), ICAM-1 4 µg/mL (ADP4050, Bio-Techne GmbH, Wiesbaden, Germany), and CXCL8/IL-8 10 µg/mL (200-08M, Peprotech, London, UK) was successfully used to induce adult leukocyte adhesion [11]. In dose-finding experiments, coating concentrations ranging from 1.0 to 20 μg/mL were tested in the novel setup with neonatal PMNs. Coating with 2 μg/mL P-selectin, 2 µg/mL ICAM-1, and 5 µg/mL IL-8 for two hours at room temperature yielded the most robust and reproducible results, and these concentrations were used for all experiments.

Leukocyte rolling and adhesion was analyzed with or without previous incubation with 10 µg/mL L-arginine (A8094, Sigma Aldrich, Germany) for 45 min on ice. The number of rolling cells passing an imaginary line across the entire chamber was counted for one minute.

Permanently (<30 s) adherent cells were counted as neutrophil adhesion per field of view (FOV) after 15 min. Images were recorded via a CCD camera system (CF8HS, Kappa, Gleichen, Germany) and were analyzed in InSpector Pro software (4.0.469, Lavision Biotec GmbH, Bielefeld, Germany).

### 2.5. Transmigration of Polymorphonuclear Leukocytes (PMNs)

Leukocyte transmigration was analyzed using transmigration chambers (Corning^®^ HTS Transwell^®^ 96 well permeable, S058.3387, Stein Labortechnik, Remchingen, Germany). Cells were pre-incubated with or without L-arginine (10 µg/mL) for 45 min. Migration was compared to buffer (HBSS, 14.025.050, Life Technologies, Carlsbad, CA, USA) alone as a negative control or along an imposed apical gradient of IL-8 (200 ng/mL) over 45 min at 37 °C, 5% CO_2_. PMNs that transmigrated were stained with fluorescent calcein (C1430, Invitrogen, Carlsbad, CA, USA) for 45 min at 37 °C, 5% CO_2_. Cells were washed twice in HBSS (+ 0.5% BSA) and lysed by hypotonic buffer (CTAB buffer, Cetyltrimethylammonium bromide, 9161.1, Carl Roth, Germany). Fluorescence intensity was quantified using a SpectraMax M2 plate reader (wavelength 485–535 nm).

### 2.6. Flow Cytometry

The expression of Arg-1 and iNOS in neutrophils was analyzed by flow cytometry. We performed red blood cell lysis on whole blood, as described previously. Cells were permeabilized using 0.1% saponin and stained with a human iNOS–FITC-conjugated antibody (NBP2-22119F, Novusbio, London, UK) and a human Arginase 1-PE conjugated antibody (678801, Biolegend, London, UK) at a concentration of 2.5 µg/mL. Measurements were performed using an LSR II cytometer (BD Biosciences, Heidelberg, Germany) and analyzed by FACS Diva (Becton Dickinson, San Jose, CA, USA) and FlowJo software version 10.1r5 (Ashland, OR, USA). The expression of Arg-1 and iNOS was compared to their respective isotype controls (mouse IgG2b, k-PE (369703, Biolegend, London, UK) and mouse IgG1, k FITC (11-4714-73, eBiocience. San Diego, CA, USA)).

### 2.7. Statistics

Statistics were performed using Prism software (version 6.01, GraphPad Software Inc., San Diego, CA, USA) and Stata (version 13, Stata Software Inc., College Station, TX, USA). Significance was set at *p* < 0.05. Data are presented as mean +SEM. For logarithmic data such as mean fluorescent intensities (MFIs) from flow cytometry, geometric means were used for statistical analysis and data presentation. Clinical and laboratory parameter of patients were compared by Student’s *t*-test and Chi-square test. Leukocyte adhesion, neutrophil transmigration, and Arg-1 and iNOS expression between groups were compared by ANOVA (one-way or two-way, wherever appropriate) followed by multiple pairwise post-hoc analysis (Tukey’s test). Linear regression and Pearson’s correlation were performed for the gestational age of preterm infants and adherence of leukocytes. Individual tests are mentioned in the respective figure legends.

## 3. Results

### 3.1. Study Population

A total of *n* = 63 infants and *n* = 17 healthy adult donors were recruited for our study from 01/2018 to 08/2019. *n* = 44 preterm infants (28 + 0 – 32 + 6 weeks of gestation) were recruited through the PRIMAL cohort, and an additional *n* = 17 term born (≥ 37 + 0 weeks of gestation) infants were included (Table 1). Reasons for preterm delivery were placental insufficiency, pre-eclampsia, hemolysis-elevated liver enzymes and low platelets (HELLP) syndrome, pathologic Doppler flow, and twin pregnancy. Children with underlying malformations or lethal conditions were excluded. The patient characteristics and laboratory parameters for all infants are shown in Table 1. As anticipated, WBC counts were lower in premature infants and more premature infants required treatment with antibiotics (89% in preterm versus 0% in term-born infants). On the contrary, all term infants were breastfed on the first day of life, whereas only 52% of the preterm population received breastmilk.

Leukocyte rolling, adhesion, and transmigration is decreased by L-arginine only in term-born infants.

A novel microflow chamber setup using Ibidi µ-Slides was designed to facilitate the testing of different experimental conditions from the same infant with a limited amount of blood and resulting PMNs (Figure 1). Dose-finding experiments revealed an optimal L-arginine concentration of 1 mg/mL. Coating with P-selectin, IL-8, and ICAM-1 induced leukocyte rolling and adhesion in preterm and term infants as well in adults, whereas the extent gradually increased with age (Figure 2). The pretreatment of leukocytes from term-born infants and adults with L-arginine significantly dampened both rolling and adherence; however, no effect was seen for preterm infants. In adults, the effect of L-arginine of leukocyte rolling was less pronounced compared to the effect on adherence. Of note, the number of adherent neutrophils reached similar levels in all groups. Next, we tested the transmigration capacity along an IL-8 gradient (Figure 3). IL-8 did not induce a significant migratory response in preterm leukocytes. Likewise, pre-incubation with L-arginine had no impact on cell migration. Leukocytes of term-born infants and adults showed both an induction of transmigration with IL-8 and a reduction after incubation with L-arginine (Figure 3).

### 3.2. The Response of L-Arginine Does Not Depend on Clinical Characteristics at Birth and Throughout Neonatal Development

A subset of preterm infants of *n* = 17 was followed up after 28 days to assess the postnatal maturation of leukocyte recruitment. As expected, leukocyte adhesion correlates with gestational age and postnatal maturation (Figure 4). Next, we asked whether the responsiveness to L-arginine treatment was dependent on clinical characteristics such as the inflammatory status. We performed subgroup analyses for preterm infants on day 1 and day 28 depending on their L-arginine responsiveness assessed by cell adhesion in microflow chamber experiments. The mean reduction of adhesion/mm^2^ upon L-arginine treatment (delta of coated − L-arginine and coated + L-arginine) served as an arbitrary cut-off to define the group of “responders” and “non-responders” (Table 2). Surprisingly, except for differences in head circumference, no other differences in clinical parameters were seen between responders and non-responders both after birth and after 28 days. Perinatal factors, the use of antibiotics, nutrition, and inflammatory parameters such as CRP or WBC counts did not show any difference between subgroups. However, the responsiveness to L-arginine depends on the baseline adhesion capacity of leukocytes (Table 2, second row/coated chambers).

### 3.3. Expression of Arginase 1 and iNOS Is not Ontogenetically Regulated

We next asked whether a lack of L-arginine response in the recruitment of leukocytes from preterm infants originated from a lack of enzymatic machinery to process the amino acid (Figure 5). Flow cytometric analysis revealed stable and reproducible levels of both Arg-1 and iNOS on PMNs across all groups. In line with the previously described similar levels of leukocyte adhesion in the L-arginine-treated PMNs of all experimental groups (Figure 2), levels of iNOS and Arg-1 were almost identical among all age groups.

## 4. Discussion

Sepsis continues to be a major cause of mortality and morbidity in preterm infants [27]. L-arginine is an important amino acid in inflammatory disease due to immunomodulatory properties. Since L-arginine levels are low in preterm infants [17,18,28,29] and are known to drop during sepsis [19], it has been suggested to supplement L-arginine among the most vulnerable preterm population. This makes sense, as L-arginine depletion inhibits macrophages, as well as T- [30,31] and NK cells [12]. However, leukocyte recruitment has been shown to be dampened by L-arginine [14], which might be detrimental in sepsis. Therefore, we asked whether the effect of L-arginine on the leukocyte recruitment cascade was altered in preterm infants compared to leukocytes from term-born infants or adults. Our data show that the effect of L-arginine on leukocyte recruitment in our ex vivo assays greatly depends on gestational age.

As expected, we found leukocyte rolling, adhesion, and transmigration to be ontogenetically regulated. Further, leukocytes underwent a similar maturation process postnatally, as shown for a subgroup of preterm infants with sequential blood draws. Pre-incubation with 10 µg/mL L-arginine was roughly equivalent to physiological L-arginine plasma concentrations in infants of around 60 µmol/L, as reported previously [29]. However, at this concentration, leukocyte recruitment of the preterm subgroup remained unaffected. Leukocyte adhesion reached equal values among all experimental groups after L-arginine incubation, suggesting a similar machinery to process L-arginine in leukocytes from all age groups. Consequently, we found levels of Arg-1 and iNOS in peripheral PMNs to be almost identical. Using transcriptomics and functional pathway analysis, Raymond et al. unveiled iNOS signaling as one of three pathways that were critically impaired in neutrophils from preterm infants compared to term-borns [32]. As iNOS is involved in pro-inflammatory activation, this might partially explain a lack of responsiveness to L-arginine supplementation despite adequate levels of iNOS in the PMNs of preterm infants. Interestingly, both leukocyte rolling and adherence were dampened by L-arginine in term-born infants and adults. The question remains of how L-arginine affects leukocyte recruitment ex vivo. It was shown that the expression of integrins such as LFA-1, Mac-1 or VLA-4 as well as L-selectin remains unchanged after exposure to L-arginine [33]. Another target could be CD162 (PSGL-1), which binds to P-selectin and is known to be ontogenetically regulated [8]. PSGL-1 mediates leukocyte rolling and interacts with L-selectin to form the PSGL-1–L-selectin complex, which signals through Src family kinases to activate LFA-1, leading to firm arrest and adherence [34]. Modulation of this complex could account for both reduction in rolling and adherence. It remains an interesting biological phenomenon that L-arginine supplementation acts as a pro-inflammatory stimulus for most immune cells, whilst it reduces the activation of neutrophils. As neutrophils are among the first cells to arrive at the site of inflammation, they might act as regulatory cells. They have been shown to locally reduce L-arginine levels through Arginase-1 activity and thereby to suppress other immune cells [35]. As such, neutrophils may be important in the micro-regulation of inflammatory responses and an important player in the checks and balances of acute inflammation.

Many immunomodulatory effects of L-arginine converge on the production of NO. NO regulates functional attributes of neutrophils such as their chemotactic, phagocytic, migratory, and apoptotic activities through cGMP-dependent and independent pathways [36]. NO mediates neutrophil extracellular traps (NET) release at the site of inflammation. Increasing arginine availability not only restores nitric oxide production but it may also mitigate tissue necrosis by modulating hypoxia, inhibit platelet activation and leukocyte adherence, and increase free radical scavenging [37,38]. In turn, L-arginine availability is partially determined by the expression of enzymes that are involved in the biosynthesis of arginine such as CPS-1, OCT, P5CR, ASS or ASL, which are suppressed in preterm neonates [39,40]. Delayed hepatic maturation might also partially account for lower levels of L-arginine in preterm infants, as the largest amount of L-arginine is synthesized in the liver. Similarly, Robinson et al. showed in preclinical NEC models that enzymes for the synthesis of citrulline and arginine were lower in preterm infants, whereas the gene expression of enzymes that catabolize arginine did not differ [20].

It is pertinent to recognize some limitations of this study. The study was designed to address the effect of leukocytes ex vivo and independent of endothelial contribution. The effect of L-arginine metabolism of the liver was also omitted. In addition, transmigration above baseline along an IL-8 gradient was not detected for the leukocytes of preterm infants, despite similar CXCR-2 levels between preterm and term infants [8]. Potentially, the assay was not sensitive enough to detect transmigration in a setting of limited cell numbers, and more sensitive methods might have been more appropriate to rule out technical limitations [41,42]. The use of L-arginine as an immune modulator in preterm infants needs to be weighed against potential adverse effects on hemodynamic stability through increased NO synthesis, which is a question of debate [43,44,45]. Even in the case of NEC, where L-arginine supplementation has been widely discussed, a recent Cochrane analysis does not give a definitive recommendation due to the lack of clinical evidence [23].

This is to our knowledge the first systematic study on the effect of L-arginine on the leukocyte recruitment cascade in infants. Our data suggest that no relevant effect of L-arginine supplementation on leukocyte recruitment is seen in preterm infants; however, ontogenetic regulation suggests anti-inflammatory effects in term-born infants by suppressing leukocyte recruitment. There are two different translational deductions, which might be drawn from our study. On one hand, L-arginine supplementation in very preterm infants may support immune responses without negatively affecting recruitment of neutrophils. On the other hand, the potential of L-arginine to inhibit neutrophils during SIRS is limited and should be weighed carefully against potential hemodynamic complications as a result of NO production. The study highlights that immunomodulation in the newborn needs to be tailored to gestational age and the maturity of the immune system.

## Figures and Tables

**Figure 1 jcm-09-02772-f001:**
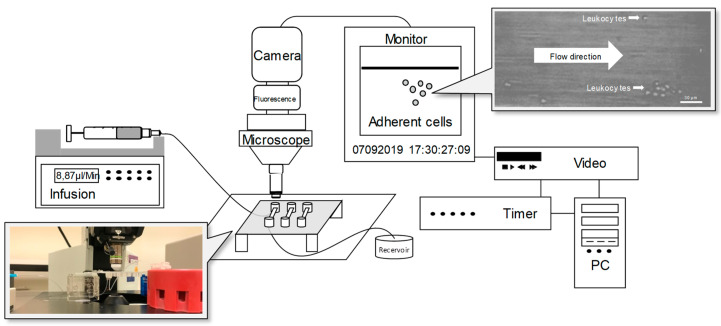
Microflow chamber setup. Neutrophil adhesion of preterm, term infants, and adults was analyzed in dynamic microflow chamber experiments. To mimic blood flow conditions, a high-precision perfusion pump with a 1 dyn/cm^2^ wall shear rate and microglass capillaries, which were coated with P-Selectin, intercellular adhesion molecule 1 (ICAM-1), and IL-8 were used. The chambers were connected via PE tubing to a 1 mL syringe containing freshly isolated neutrophils. Permanent adhered cells were counted as neutrophil adhesion per field of view (FOV) after 15 min.

**Figure 2 jcm-09-02772-f002:**
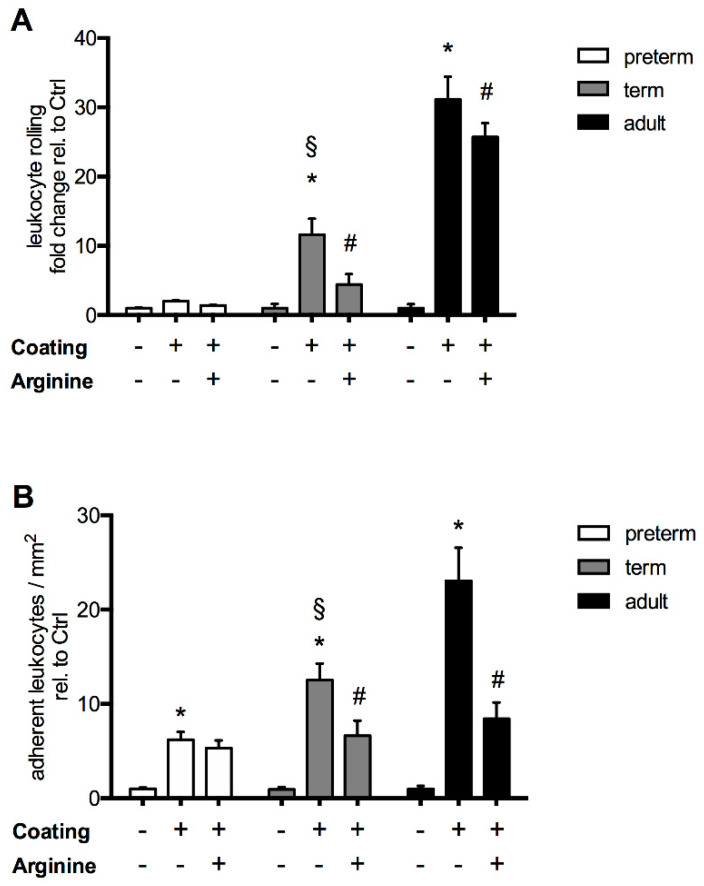
Leukocyte rolling and adhesion is decreased by L-arginine in term infants. Leukocyte rolling (**A**) and adhesion (**B**) of preterm and term infants and adults was analyzed in dynamic microflow chamber experiments. Neutrophil rolling and adhesion upon P-selectin, IL-8, and ICAM-1 coating was compared between the various age groups and to respective uncoated controls or cells that were incubated in 10 µg/mL L-arginine for 45 min. Results are presented as mean + SEM from at least 5 separate individuals/experiments per group. Statistical analysis was performed by two-way ANOVA and Tukey’s post hoc test. Significance was set at *p* < 0.05. * indicates significance compared to uncoated controls, # indicates significant downregulation compared to coated chambers, and § indicates significant differences to both preterm and adult coated chambers.

**Figure 3 jcm-09-02772-f003:**
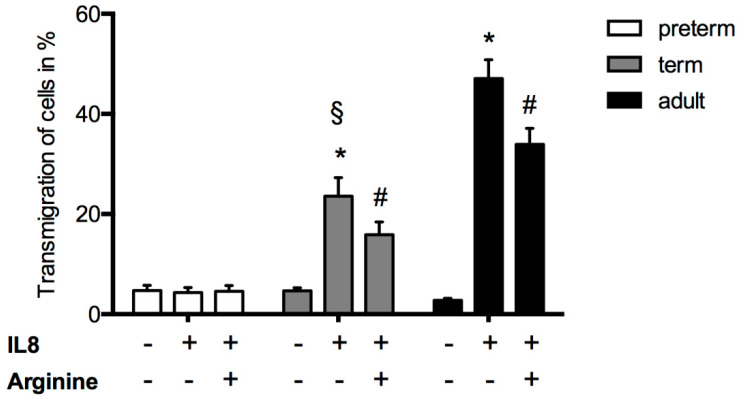
Leukocyte transmigration is decreased by L-arginine in term infants. Leukocyte transmigration of preterm and term infants and adults was analyzed in transmigration assays in the presence or absence of an IL-8 gradient and or incubation with 10 µg/mL L-arginine for 45 min. Results are presented as mean +  SEM from at least 5 separate individuals/experiments per group. Statistical analysis was performed by two-way ANOVA and Tukey’s post hoc analysis. Significance was set at *p* < 0.05. * indicates significance compared to controls (IL-8 (−), arginine (−)), # indicates significant downregulation of transmigration compared to IL-8 stimulated cells (IL-8 (+), arginine (−)) and § indicates significant differences to both preterm and adult IL-8 stimulated cells (IL-8 (+), arginine (−)).

**Figure 4 jcm-09-02772-f004:**
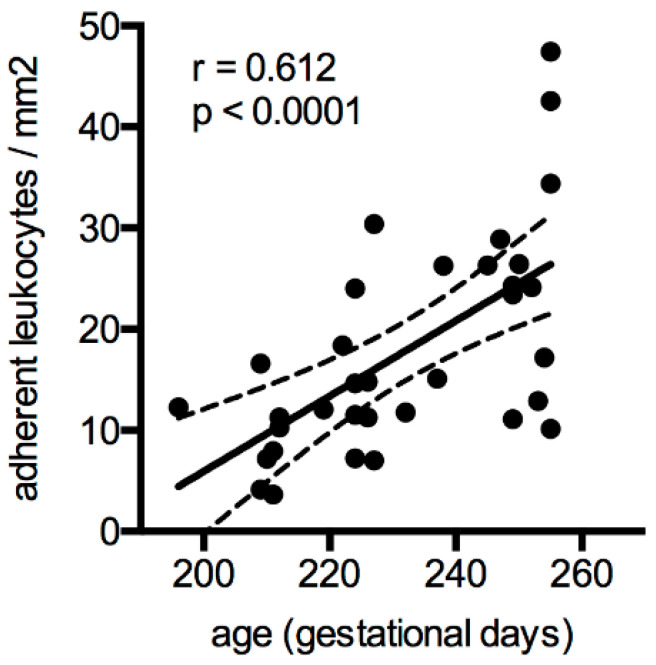
Leukocyte adhesion correlates with postnatal development. Leukocyte adherence from microflow chamber experiments from the *n* = 17 preterm infants at day 1 and day 28 of life was correlated to gestational age (in days). Linear regression showed a significant correlation (Pearson r = 0.612). Data are presented as individual data points with linear regression (solid line) and 95% confidence intervals (dashed line).

**Figure 5 jcm-09-02772-f005:**
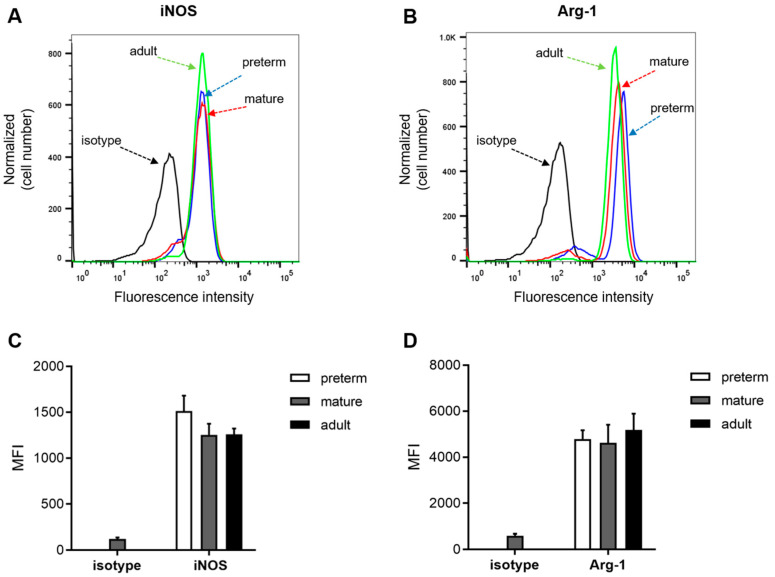
Expression of Arginase 1 (Arg-1) and inducible nitric oxide synthase (iNOS) is not ontogenetically regulated. Expression of Arg-1 and iNOS on peripheral polymorphonuclear leukocytes (PMNs) were compared by flow cytometry. Representative histograms are shown for both Arg-1 (**A**) and iNOS (**B**). Quantification of mean fluorescence intensity (MFI) is shown for both Arg-1 (**C**) and iNOS (**D**). Data presented as mean + SEM from at least 5 separate experiments. No statistical differences were found by one-way ANOVA with Tukey’s post hoc analysis. Significance was set at *p* < 0.05.

**Table 1 jcm-09-02772-t001:** Basic and clinical characteristics of study population.

Clinical Characteristics	Preterm Infants	Term Infants	*p*-Value
	Day 1	Day 1	
	*n* = 44	*n* = 17	
GA in weeks + days, mean ± SD	30 + 6 ± 1 + 3	38 + 3 ± 1 + 5	**<0.0001**
5 min APGAR, mean ± SD	8.0 ± 0.9	9.4 ± 0.9	**<0.0001**
10 min APGAR, mean ± SD	8.6 ± 0.7	9.8 ± 0.6	**<0.0001**
Male gender, *n* (%)	29 (66)	11 (65)	0.714
Weight in g, mean ± SD	1569 ± 355	3045 ± 464	**<0.0001**
Head circumference in cm, mean ± SD	28.8 ± 1.5	34.4 ± 1.5	**<0.0001**
Body length in cm, mean ± SD	41 ± 3	51 ± 3	**<0.0001**
Nutrition, *n* (%)			
Breast milk exclusively	23 (52)	17 (100)	**0.0004**
Formula supplementation	21(48)	0	
PROM, *n* (%)	10 (23)	0	**0.032**
Antibiotics, *n* (%)	39 (89)	0	**<0.0001**
Laboratory parameters on admission			
Arterial pH, mean ± SD	7.3 ± 0.1	7.3 ± 0.1	0.483
CRP > 2 mg/L, *n* (%)	1 (2)	2 (12)	0.124
WBC count/nL, mean ± SD	10.1 ± 3.9	13.7 ± 4.9	**0.004**
I/T ratio > 0.25, *n* (%)	2 (4)	0	0.371

Categorical variables are reported as *n* (%) and continuous variables as mean with SD. Group comparison were performed using Chi-square for categorical variables and by Student’s *t*-test for continuous variables. *p*-values < 0.05 were considered statistically significant and are marked in bold. GA: gestational age; APGAR: appearance, pulse, grimace, activity und respiration score; PROM: Premature rupture of membranes; CRP: C-reactive protein; WBC count: white blood cell count; I/T ratio: immature/total neutrophil ratio.

**Table 2 jcm-09-02772-t002:** Basic and clinical characteristics of preterm infants depending on L-arginine response at birth and on day 28. Subgroups are divided by the mean reduction in leukocyte adherence by L-arginine (L-arginine response) for each time point (day 1, 28). Categorical variables are reported as *n* (%) and continuous variables are reported as mean with SD. Group comparison were performed using Chi-square for categorical variables and by Student’s *t*-test for continuous variables. Adherence from microflow chamber experiments was compared by two-way ANOVA and Tukey’s post hoc analysis. *p*-values < 0.05 were considered statistically significant and are marked in bold.

Flow Chamber, Adherence/mm^2^, Mean ± SD	Preterm Infants Day 1 *n* = 17			Preterm Infants Day 28 *n* = 17		
	L-Arginine Responder *n* = 9	L-Arginine Non-Responder *n* = 8	*p*-Value	L-Arginine Responder *n* = 8	L-Arginine Non-Responder *n* = 9	*p*-Value
Uncoated	4.3 ± 2.7	2.0 ± 1.9	0.990	2.1 ± 1.6	3.0 ± 2.1	0.999
Coated − L-arginin	33.5 ± 9.4	14.3 ± 8.7	**0.0001**	36.7 ± 13.4	17.2 ± 12.0	**0.001**
Coated + L-arginin	18.3 ± 9.6	16.4 ± 10.1	0.996	11.3 ± 8.8	15.5 ± 10.7	0.938
Clinical characteristics						
GA in weeks + days, mean ± SD	216.4 ± 9.8	215.0 ± 9.1	0.759	222.1 ± 5.5	216.3 ± 9.6	0.154
APGAR 5′, mean ± SD	6.4 ± 1.7	6.0 ± 1.6	0.947	8.4 ± 1.1	8.1 ± 0.8	0.565
APGAR 10′, mean ± SD	8.6 ± 0.9	8.5 ± 0.8	0.892	8.9 ± 0.6	8.8 ± 0.7	0.764
Male gender, *n* (%)	6 (66.7)	6 (75.0)	0.707	4 (50.0)	7 (77.8)	0.232
Weight in g, mean ± SD	1584.4 ± 367.9	1673.6 ± 173.5	0.541	2465.6 ± 263.8	2452.2 ± 315.9	0.926
Head circumference in cm, mean ± SD	29.1 ± 1.7	28.9 ± 0.8	0.751	31.5 ± 1.5	33.3 ± 1.4	**0.020**
Body length in cm, mean ± SD	41.4 ± 2.8	41.0 ± 2.3	0.735	45.0 ± 1.39	46.2 ± 2.6	0.247
Breast milk feeds, *n* (%)	9 (100.0)	7 (87.5)	0.274	8 (100.0.)	7 (77.8)	0.156
PROM, *n* (%)	2 (22.2)	2 (25.0)	0.893	3 (37.5)	1 (11.1)	0.200
Antibiotics, *n* (%)	8 (89)	7 (88)	0.929	2 (25)	2 (22)	0.893
Birth mode, *n* (%)						
Vaginal spontanous	2 (22.2)	2 (25.0)	0.893	2 (25.0)	1 (11.1)	0.453
Caesarean Section	7 (77.8)	6 (75.0)		6 (75.0)	8 (88.9)	
Laboratory parameters						
Arterial pH, mean ± SD	7.3 ± 0.1	7.3 ± 0.1	0.954	7.3 ± 0.1	7.3 ± 0.1	0.203
CRP > 2 mg/L, *n* (%)	0	0		3 (37.5)	4 (44.4)	0.772
WBC count/nL, mean ± SD	11.4 ± 4.2	10.5 ± 5.0	0.573	9.3 ± 1.9	11.0 ± 2.7	0.159
I/T ratio > 0.25, *n* (%)	0	0		0	0	

GA: gestational age; APGAR: appearance, pulse, grimace, activity und respiration score; PROM: Premature rupture of membranes; CRP: C-reactive protein; WBC count: white blood cell count; I/T ratio: immature/total neutrophil ratio.

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
