# Peer review of "L-Arginine Modulates Neonatal Leukocyte Recruitment in a Gestational Age-Dependent Manner"

_jcm, 2020, doi:10.3390/jcm9092772_

Round 1
Reviewer 1 Report
Why preterm infants have higher rates of infection than term infants are not well understood. L-arginine is a precursor for NO and NO has important functions in promoting vasodilation and inhibiting neutrophil adhesion. Since preterm infants have lower levels of L-arginine, this study investigates how arginine supplementation affects adhesion and transmigration of leukocytes isolated from preterm, term and adult humans. The authors found that L-arginine dampens leukocyte adhesion in term and adult leukocytes but not in preterm leukocytes. However, leukocytes collected from the same preterm infants 28 days later had higher rates of adhesion. Age-related changes in adhesion and responsiveness to L-arginine were not related to expression of iNOS or Arg-1. The findings are important because they provide new evidence that L-arginine supplementation may not be beneficial to newborn preterm infants because their leukocytes are immunologically immature. The study design is appropriate, appears reasonably powered, and the findings are novel and significant. However, impact of the findings is dampened by confusion about data presented in Table 2.
Major Concern:
- It is not clear why Table 2 shows that there are two groups of preterm infants at 1 day and at 28 days that differ based upon their responsiveness to arginine. This does not seem consistent with data presented in Figures 1-3 indicating that adhesiveness of leukocytes from all preterm infants failed to respond to arginine. Data in figure 4 shows adhesive properties of 1 day old preterm infants increases by day 28. But, none of the figures separate the preterm infants into two groups as shown in Table 2. Please clarify.
- Related to concern 1, it would help if figure 4 was a line graph showing adhesive property of each individual between day 1 and day 28. This would help to show how many increased, remained same, or declined between 1 and 28 days.
- Figure 2 shows L-arginine reduces adhesiveness in term and adult but not preterm leukocytes. But it is interesting to see that L-arginine reduced adhesiveness to the level seen in preterm leukocytes, suggesting that L-arginine was affecting whatever change took place as leukocytes matured with age. Theoretically, increased adhesiveness is related to signaling afforded by interacting with the coated material on the plates, namely P-selectin, IL-8, and/or ICAM-1. The response to L-arginine might be identified by investigating whether expression of their receptors change with age or testing whether arginine affects adhesive properties of adult cells plated on one, two, or three of these molecules.
Minor Comments
- Line 51 would benefit by adding references supporting the statement that septic preterm infants have less L-arginine that the low levels normal seen just with preterm birth.
- Line 207, the word assess is missing an s.
- Line 228 referring to Table 2 says the responsiveness to L-arginine depends on the baseline adhesion capacity. Would that be to the plastic or the coating?
Author Response
Comments and Suggestions for Authors
Why preterm infants have higher rates of infection than term infants are not well understood. L-arginine is a precursor for NO and NO has important functions in promoting vasodilation and inhibiting neutrophil adhesion. Since preterm infants have lower levels of L-arginine, this study investigates how arginine supplementation affects adhesion and transmigration of leukocytes isolated from preterm, term and adult humans. The authors found that L-arginine dampens leukocyte adhesion in term and adult leukocytes but not in preterm leukocytes. However, leukocytes collected from the same preterm infants 28 days later had higher rates of adhesion. Age-related changes in adhesion and responsiveness to L-arginine were not related to expression of iNOS or Arg-1. The findings are important because they provide new evidence that L-arginine supplementation may not be beneficial to newborn preterm infants because their leukocytes are immunologically immature. The study design is appropriate, appears reasonably powered, and the findings are novel and significant. However, impact of the findings is dampened by confusion about data presented in Table 2.
Major Concern:
- It is not clear why Table 2 shows that there are two groups of preterm infants at 1 day and at 28 days that differ based upon their responsiveness to arginine. This does not seem consistent with data presented in Figures 1-3 indicating that adhesiveness of leukocytes from all preterm infants failed to respond to arginine. Data in figure 4 shows adhesive properties of 1 day old preterm infants increases by day 28. But, none of the figures separate the preterm infants into two groups as shown in Table 2. Please clarify.
- We agree that Table 2 is somewhat different from the other data shown in this manuscript. The aim was to delineate whether there were differences in clinical characteristics from leukocytes of infants with low response to L-arginine (small delta between coated-arginine and coated+arginine) compared to the ones with a higher response. Even if there was no net effect of L-arginine incubation on adhesion in preterm infants, some individuals showed a response, whilst others did not. By displaying the data like this, two deductions stand out:
- The absolute number of adherent cells after arginine treatment is very similar between the low and high-response group as well as between day 1 and 28 (which is in line with Figure 2)
- We did not identify a specific clinical setting that could explain different responses to L-arginine in our study. We were specifically looking for differences related to nutrition or infections.
- We have specified that the cut-off is arbitrary to highlight, that these are not two distinct populations. If the reviewer feels this table rather confuses the reader, we would also consider removing it from the manuscript.
- Related to concern 1, it would help if figure 4 was a line graph showing adhesive property of each individual between day 1 and day 28. This would help to show how many increased, remained same, or declined between 1 and 28 days.
- Figure 4 shows baseline data of leukocyte adherence at birth and on day 28 for 17 infants. Please find a line graph attached as suggested. Only 4 infants actually show decreased leucocyte adhesion after the first month of life. We could include this in the manuscript if wanted, however we feel it might distract from the main point we were trying to make with this figure (maturation of leukocyte function/adhesion takes place similarly ex utero).
(graph in attached pdf)
- Figure 2 shows L-arginine reduces adhesiveness in term and adult but not preterm leukocytes. But it is interesting to see that L-arginine reduced adhesiveness to the level seen in preterm leukocytes, suggesting that L-arginine was affecting whatever change took place as leukocytes matured with age. Theoretically, increased adhesiveness is related to signaling afforded by interacting with the coated material on the plates, namely P-selectin, IL-8, and/or ICAM-1. The response to L-arginine might be identified by investigating whether expression of their receptors change with age or testing whether arginine affects adhesive properties of adult cells plated on one, two, or three of these molecules.
- We agree that studying the individual components of the leukocyte recruitment cascade in more detail could yield information on where exactly L-arginine may exert its inhibitory effects. This is in line with the questions of reviewer #2 regarding Figure 1 and additional data on rolling. We (among many others) have previously looked into the contribution of individual mediators to the recruitment of leukocytes in various gestational stages and in adults. Please find some unpublished data on the effect of ICAM-1-deficient coating in flow chamber experiments below as an example. In the current study, we did not use cord blood samples but peripheral blood samples from preterm infants, which makes additional studies difficult and time consuming.
(graph in attached pdf)
Minor Comments
- Line 51 would benefit by adding references supporting the statement that septic preterm infants have less L-arginine that the low levels normal seen just with preterm birth.
- Badurdeen et al have reviewed L-arginine depletion in serious infection in preterm infants and it seems likely that septic preterm infants show lower arginine levels, akin to preterm infants with NEC (Badurdeen, 2015). However, we have restructured the introduction as suggested by reviewer #2 and removed this section.
- Line 207, the word assess is missing an s.
- Line 228 referring to Table 2 says the responsiveness to L-arginine depends on the baseline adhesion capacity. Would that be to the plastic or the coating?
- We are referring to coated chambers and have specified this in the manuscript.

Reviewer 2 Report
Fitterer et al. investigated the effect of L-arginine on leukocyte adhesion and transmigration using neutrophils from preterm and term neonates, as well as neutrophils isolated from adult donors. In addition, they studied expression levels of arginase 1 and iNOS depending on gestational age on peripheral human neutrophils by flow cytometry. The methods are well described and most of them chosen properly.
It was shown before, that L-Arginine has an anti-inflammatory effect in vivo and that it reduces leukocyte recruitment (Dhananjay K. Kaul, Blood, 2005). The novel part of the data is the effect of L-Arginine on neutrophil adhesion (by using in vitro flow chambers the authors were able to exclusively focus on neutrophils excluding any effect of L-arginine on the endothelium) and transmigration depending on gestational age. The approach is interesting.
General points: Could the authors maybe provide a clear(er) explanation of the objective and conclusion of their study? If sepsis is an “arginine deprived state” and L-arginine dampens immune responses, why should L-arginine be used as a supplementation in preterm or term neonates (if they in general have reduced immune response)? Would the authors suggest to use L-arginine supplementation to dampen overwhelming immune responses driven by neutrophils?
The authors claim, that L-arginine levels are 50% lover in extremely premature infants (Wilson et al. 2014). However, within the cited paper Wilson et al. show that preterm infants have higher levels of L-arginine. “Levels of several amino acids increased with higher degrees of prematurity. These findings were most evident in arginine and valine levels, which were more than 50% higher in extremely premature infants compared with those in term infants”. This finding would even fit better with the results presented in this study, as L-arginine reduces leukocyte adhesion and premature infants also display reduced adhesion in flow chamber assays as shown here and in previous studies.
Figure 1: does the reduction in adhesion in the presence of L-arginine go along with an increase in rolling or do the cells in general interact less in the presence of L-arginine? (is also rolling decreased or not affected)? What do the authors think could be the reason for the reduction in adhesion if the cells are pre-incubated with L-arginine (reduced beta2 integrine activation)? If rolling is also reduced: could L-arginine alter PSGL-1 expression levels? I think it would be very important to include possible explanations within the discussion.
Figure 2: neutrophils from preterm infants do not respond to the chemokine CXCL-8 at all. However, CXCR2 levels were shown to be equal between preterm and term neonates and adults (Nussbaum et al. 2013). Could the authors discuss possible reasons for this complete unresponsiveness of neutrophils from extreme premature infants?
Figure 5: iNOS is a cytoplasmic enzyme. Why did the authors do FACS analyses and not confocal staining or even better Western blot analyses? Did the authors do intracellular FACS staining? In my view, extracellular FACS staining of iNOS does not allow any conclusion on iNOS expression on neutrophils.
In addition, iNOS is known to be upregulated upon activation by diverse cytokines (e.g. TNF-alpha) and by TLR4 engagement. Is there anything known if iNOS upregulation is defective in preterm neonates? Is it possible, that leukocyte adhesion and transmigration (events that go along with stimulation of neutrophils) are not affected by the addition of L-arginine because iNOS upregulation is not taking place in preterm neutrophils? This issue may be beyond of the scope of the study, but maybe the authors can comment on that and include some thoughts in their discussion.
Author Response
Please also see attached pdf
Comments and Suggestions for Authors
Fitterer et al. investigated the effect of L-arginine on leukocyte adhesion and transmigration using neutrophils from preterm and term neonates, as well as neutrophils isolated from adult donors. In addition, they studied expression levels of arginase 1 and iNOS depending on gestational age on peripheral human neutrophils by flow cytometry. The methods are well described and most of them chosen properly.
It was shown before, that L-Arginine has an anti-inflammatory effect in vivo and that it reduces leukocyte recruitment (Dhananjay K. Kaul, Blood, 2005). The novel part of the data is the effect of L-Arginine on neutrophil adhesion (by using in vitro flow chambers the authors were able to exclusively focus on neutrophils excluding any effect of L-arginine on the endothelium) and transmigration depending on gestational age. The approach is interesting.
General points: Could the authors maybe provide a clear(er) explanation of the objective and conclusion of their study? If sepsis is an “arginine deprived state” and L-arginine dampens immune responses, why should L-arginine be used as a supplementation in preterm or term neonates (if they in general have reduced immune response)? Would the authors suggest to use L-arginine supplementation to dampen overwhelming immune responses driven by neutrophils?
- We agree, that the objective was not entirely clear, especially as we maybe over-simplified introduction. Newborn infants and especially preterm infants show impaired activation of immune responses but also fail to adequately resolve inflammation. Therefore, different stages of sepsis may require different modulators. Ultimately most infants on the NICU, who suffer fatal consequences of sepsis undergo dysregulated hyper-inflammation (SIRS). Therefore, suppressing leukocyte recruitment may be one of many ways to resolve SIRS in late stages of sepsis. However, at early stages, pro-inflammatory activation is more important. Interestingly, L-arginine depletion generally reduces immune cell function and supplementation has pro-inflammatory effects. For example it was shown in preclinical murine models, that arginine limitation dampens TLR-4 signaling in macrophages and supplementation of arginine can restore TNF- α production in response to LPS (Mieulet, 2010). Other studies in human cells have found that arginine depletion through neutrophil arginase secretion suppresses T-cell function, again, restoring arginine levels had a pro-inflammatory effect (Munder, 2006). By a similar mechanism, human NK cells are suppressed by PMN arginase activity and resulting arginine depletion (Oberlies, 2009). On the other hand arginine has been identified as a suppressor of leukocyte recruitment as mentioned by the reviewer. Also, it has been shown that L-arginine downregulates expression of ICAM-1 in human endothelial cells, thereby affecting leukocyte adherence (Adams, 1997).
This opposite effect of L-arginine on immune cells is surprising, which is why we took an isolated look at neutrophils in leukocyte recruitment (independent of other immune cells or modulators). We have highlighted this background in more detail in the introduction, sharpened the focus of the research question and revisited the discussion.
The authors claim, that L-arginine levels are 50% lover in extremely premature infants (Wilson et al. 2014). However, within the cited paper Wilson et al. show that preterm infants have higher levels of L-arginine. “Levels of several amino acids increased with higher degrees of prematurity. These findings were most evident in arginine and valine levels, which were more than 50% higher in extremely premature infants compared with those in term infants”. This finding would even fit better with the results presented in this study, as L-arginine reduces leukocyte adhesion and premature infants also display reduced adhesion in flow chamber assays as shown here and in previous studies.
- Thank you for the very thorough revision – indeed we cited the paper entirely opposite to what it says – please excuse this lack of scientific rigor… I am not sure as to why arginine levels were much higher in extremely preterm infants in this particular study. I can only speculate that it must be associated with the sampling time-point as all measurements were from the newborn screening after birth. Despite this work, we are convinced that that L-arginine levels are lower in premature infants, please see the very comprehensive review by Wu et al (Wu, 2004). In clinical practice L-arginine is supplemented in all infants <28 weeks at our institution (which is why we only sampled blood from infants >28 weeks of gestation). However, whilst addressing the reviewer’s comments, we have come to the conclusion that the absolute levels of L-arginine in preterm infants are not integral to this work. In the ex vivo setting, leukocytes from all groups are incubated with the same concentration of L-arginine and still show very different effects. We have therefore removed the citation from Wilson et al. and shifted the discussion more towards how L-arginine and neutrophils may interact and what the biological implications of this interaction may be.
Figure 1: does the reduction in adhesion in the presence of L-arginine go along with an increase in rolling or do the cells in general interact less in the presence of L-arginine? (is also rolling decreased or not affected)? What do the authors think could be the reason for the reduction in adhesion if the cells are pre-incubated with L-arginine (reduced beta2 integrine activation)? If rolling is also reduced: could L-arginine alter PSGL-1 expression levels? I think it would be very important to include possible explanations within the discussion.
- We have added rolling data from our experiments (Figure 2) and indeed there seems to be less interaction in the presence of L-arginine in term born infants and adults. Similar to our adherence data there is no significant change in preterm infants. PSGL-1 would be a potential target explaining the effects of L-arginine on both rolling and adherence in our system. Either, PSGL-1 expression could be altered, or some other element of the PSGL-1-L-selectin signaling complex as described by Stadtmann et al., could be affected. The complex has been shown to both regulate binding to P-selectin and to regulate the extension of LFA-1 in firm arrest through L-selectin interaction (Stadtmann, 2013). An older study looked at granulocytes after L-arginine supplementation in humans and did not find altered expression of LFA-1, Mac-1, VLA-4 or L-selection (Blum, 2000). We have edited the discussion accordingly.
Figure 2: neutrophils from preterm infants do not respond to the chemokine CXCL-8 at all. However, CXCR2 levels were shown to be equal between preterm and term neonates and adults (Nussbaum et al. 2013). Could the authors discuss possible reasons for this complete unresponsiveness of neutrophils from extreme premature infants?
- I believe the lack of transmigration in our system for preterm leukocytes is not due to impaired CXCL-8 signaling, but rather a technical limitation. The transwell transmigration assay requires substantial numbers of cells an we were especially limited in the very preterm infant group. Potentially, total transmigration may have been under the detection level of the assay. For future projects we will establish microfluidic assays with whole blood samples as have been established recently (Jones, 2016; Hoang, 2013)
Figure 5: iNOS is a cytoplasmic enzyme. Why did the authors do FACS analyses and not confocal staining or even better Western blot analyses? Did the authors do intracellular FACS staining? In my view, extracellular FACS staining of iNOS does not allow any conclusion on iNOS expression on neutrophils.
- We permeabilized cells prior to staining and FACS analysis – we are afraid we did not mention this adequately in the methods section and have added this information. We have tried using Westerns, however we had insufficient cell numbers, as flow analyses were performed on the same samples used for functional tests.
In addition, iNOS is known to be upregulated upon activation by diverse cytokines (e.g. TNF-alpha) and by TLR4 engagement. Is there anything known if iNOS upregulation is defective in preterm neonates? Is it possible, that leukocyte adhesion and transmigration (events that go along with stimulation of neutrophils) are not affected by the addition of L-arginine because iNOS upregulation is not taking place in preterm neutrophils? This issue may be beyond of the scope of the study, but maybe the authors can comment on that and include some thoughts in their discussion.
- We performed Flow-cytometry for iNOS on LPS-stimulated neutrophils as part of this project on two individual preterm infants. Indeed, preterm leukocytes failed to upregulate iNOS, however we have not looked into this systematically and have not compared expression to term infants or adults. However, throughout this manuscript we did not stimulate cells (beyond handling) and in those cells we found no changes in iNOS expression (Figure 5). Raymond et al performed transcriptome analysis of preterm versus term and adult neutrophils (Raymond, 2018). Functional pathway analysis unveiled iNOS signaling as one of three pathways that were critically impaired in preterm infants compared to term borns. It is therefore possible that iNOS expression or signaling are dysfunctional in preterm neutrophils which would explain the lack of responsiveness to L-arginine.
- On a systemic level arginase is the more relevant metabolizer of L-arginine in sepsis though. Weiss et al. measured L-arginine levels in pediatric septic patients and found (despite decreased L-arginine levels) decreased ratio of arginine:ornithine suggesting increased arginase activity, but no difference in citrulline:arginine ratio, which is an indirect measure of NOS synthase activity (Weiss, 2012).
Bibliography
Adams MR, Jessup W, Hailstones D, Celermajer DS. L-arginine reduces human monocyte adhesion to vascular endothelium and endothelial expression of cell adhesion molecules. Circulation. 1997;95(3):662-668. doi:10.1161/01.cir.95.3.662
Badurdeen S, Mulongo M, Berkley JA. Arginine depletion increases susceptibility to serious infections in preterm newborns. Pediatr Res. 2015;77(2):290-297. doi:10.1038/pr.2014.177
Blum A, Hathaway L, Mincemoyer R, et al. Oral L-arginine in patients with coronary artery disease on medical management. Circulation. 2000;101(18):2160-2164. doi:10.1161/01.cir.101.18.2160
Hoang AN, Jones CN, Dimisko L, et al. Measuring neutrophil speed and directionality during chemotaxis, directly from a droplet of whole blood. Technology (Singap World Sci). 2013;1(1):49. doi:10.1142/S2339547813500040
Jones CN, Hoang AN, Martel JM, et al. Microfluidic assay for precise measurements of mouse, rat, and human neutrophil chemotaxis in whole-blood droplets. J Leukoc Biol. 2016;100(1):241-247. doi:10.1189/jlb.5TA0715-310RR
Mieulet V, Yan L, Choisy C, et al. TPL-2-mediated activation of MAPK downstream of TLR4 signaling is coupled to arginine availability. Sci Signal. 2010;3(135):ra61. Published 2010 Aug 17. doi:10.1126/scisignal.2000934
Munder M, Schneider H, Luckner C, et al. Suppression of T-cell functions by human granulocyte arginase. Blood. 2006;108(5):1627-1634. doi:10.1182/blood-2006-11-010389
Oberlies J, Watzl C, Giese T, et al. Regulation of NK cell function by human granulocyte arginase. J Immunol. 2009;182(9):5259-5267. doi:10.4049/jimmunol.0803523
Raymond SL, Mathias BJ, Murphy TJ, et al. Neutrophil chemotaxis and transcriptomics in term and preterm neonates. Transl Res. 2017;190:4-15. doi:10.1016/j.trsl.2017.08.003
Stadtmann A, Germena G, Block H, et al. The PSGL-1-L-selectin signaling complex regulates neutrophil adhesion under flow. J Exp Med. 2013;210(11):2171-2180. doi:10.1084/jem.20130664
Weiss SL, Haymond S, Ralay Ranaivo H, et al. Evaluation of asymmetric dimethylarginine, arginine, and carnitine metabolism in pediatric sepsis. Pediatr Crit Care Med. 2012;13(4):e210-e218. doi:10.1097/PCC.0b013e318238b5cd
Wu G, Jaeger LA, Bazer FW, Rhoads JM. Arginine deficiency in preterm infants: biochemical mechanisms and nutritional implications. J Nutr Biochem. 2004;15(8):442-451. doi:10.1016/j.jnutbio.2003.11.010

Round 2
Reviewer 2 Report
Dear authors,
thanks for taking into account my comments and suggestions.
minor points: typing error line 45